# Identification of exacerbation risk in patients with liver dysfunction using machine learning algorithms

**Junfeng Peng[1]◉, Mi Zhou[2]◉, Chuan Chen[1], Xiaohua Xie[1], Ching-Hsing Luo[1]***

**1** School of Data and Computer Science, Sun Yat-sen University, Guangzhou, Guangdong, China, **2** The Third Affiliated Hospital, Sun Yat-sen University, Guangzhou, Guangzhou, China

◉ These authors contributed equally to this work.
* luojinx5@mail.sysu.edu.cn

**Data Availability Statement:** All relevant data are within the paper and its Supporting Information files.

**Funding:** This work was supported by Sun Yat-sen University, China, under Scientific Initiation Project

## Abstract

The prediction of the liver failure (LF) and its proper diagnosis would lead to a reduction in the complications of the disease and prevents the progress of the disease. To improve the treatment of LF patients and reduce the cost of treatment, we build a machine learning model to forecast whether a patient would deteriorate after admission to the hospital. First, a total of 348 LF patients were included from May 2011 to March 2018 retrospectively in this study. Then, 15 key clinical indicators are selected as the input of the machine learning algorithm. Finally, machine learning and the Model for End-Stage Liver Disease (MELD) are used to forecast the LF deterioration. The area under the receiver operating characteristic (AUC) of MELD, GLMs, CART, SVM and NNET with 10 fold-cross validation was 0.670, 0.554, 0.794, 0.853 and 0.912 respectively. Additionally, the accuracy of MELD, GLMs, CART, SVM and NNET was 0.669, 0.456, 0.794, 0.853 and 0.912. The predictive performance of the developed machine model execept the GLMs exceeds the classic MELD model. The machine learning method could support the physicians to trigger the initiation of timely treatment for the LD patients.

## Introduction

The liver is an important and complex organ of the human body [1]. The liver performs a variety of important physiological functions such as excretory function, secretory function and detoxification [2]. The liver is exposed to many diseases such as hepatitis, fatty liver, cancer, drug damage [3]. Recent research demonstrated that liver disease (LF) could progress into fatal chronic diseases. Some liver disease such as viral hepatitis would lead to permanent liver failure [4].

The Model for End-Stage Liver Disease (MELD) is the most popular indicator for distinguishing different stages of the liver condition [5]. MELD includes four factors: creatine (Cr), total bilirubin (TBIL), prothrombin standardized ratio (INR) and etiology. The units of Cr and TBIL is mg/dL. The Etiology is defined below: bile siltation and alcohol is 0, and the rest is 1. The units of creatinine and total bilirubin data are converted to standard units (mg/dL).

No.67000-18821109 for High-level Experts. No
competing financial interests exist.

**Competing interests:** The authors have declared
that no competing interests exist.

MELD is represented as Eq (1) (model 1):

$$r = 11.2ln(INR) + 9.6In(Cr) + 3.8In(TBIL) + 6.4(Etiology) \quad (1)$$

However, the liver is related to the other important organs, such as heart, brain, lungs, kidneys, stomach, pancreas, intestines and endocrine organs. Thus, it is difficult to identify the onset of liver failure with some clinical symptoms or indicators [6]. Hence, a more precise and fit-for-purpose indicator or model is needed [7].

In recent years, artificial intelligence and machine learning methods have been used to diagnose liver disease. Jose et al adopted an artificial neural network (ANN)) as a diagnosis support tool to assist the clinical decision-making. The ANN achieved 95.4% accuracy on the test set [8]. Nevertheless, ANN, as a black box model, is poorly clinically interpretable. Further, Wu et al employed multiple machine learning methods (random forest (RF), naive bayes (NB), ANN and logistic regression (LR)) to classify the high-risk patients with fatty liver disease. The area under the receiver operating characteristic (AUC) of RF, NB, ANN, and LR with 10 fold-cross validation was 0.925, 0.888, 0.895, and 0.854 respectively [9]. The results showed that machine learning methods may have great potential in predicting the specific liver diseases.

Vijayarani et al took advantage of naïve bayesian network and support vector machine on the Indian Liver Patient Dataset (ILPD) and obtained the accuracy of 55% and 76%, respectively [10]. Olaniy et al. used neural network and radial function on BUPD dataset and achieved the accuracy of 63% and 70%, for each of the proposed methods, respectively [11]. Vahid et al proposed a hybrid algorithm using whale optimization and k-nearest neighbor algorithm on the dataset from medical research limited, India (BUPA) and ILPD. The proposed hybrid algorithm gained the accuracy of 81.24% and 91.28%, respectively [12]. However, they did not include the important clinical significance indicators, such as sodium, creatinine, international normalized prothrombin ratio, and absolute value of neutrophils. Studies showed that the high neutlyte granulocytes count is closely related to the short-term mortality rate of one month in patients with cirrhosis, and is positively correlated with complications associated with the failure of cirrhosis [13]. Further, MELD score combined with serum sodium may improve the short-term prognosis in patients with cirrhosis [14].

Thus, it is of great clinical significance to establish a predictive model for the LF containing more clinical indicators. We employed multiple machine learning methods to predict the deterioration risk of LF with 15 key clinical indicators. Specially, the proposed classification and regression tree (CART) [15] achieved good prediction performance and got high interpretation among the five models. CART showed 79.41% (54 out of 68 participants) accuracy with 95% Confidence Interval (CI): (0.6788, 0.8826). While, MELD gained the 67% accuracy. Experimental results showed that the machine learning method performs well to predict the early LF exacerbation.

## Materials and methods

### Data acquisition

The research was specifically approved by the Institutional Review Board (IRB) of the the Third Affiliated Hospital, Sun Yat-sen University (TAHSYU) with the protocol #[2019]-02-334-01. All patient information is anonymized and de-identified before analysis.

We conducted a retrospective study and obtained 348 EHRs with 15 clinical indicators from the hepatobiliary unit database of TAHSYU, a major Chinese large-scale Medical Center. Table 1 shows the basic condition of the LF patients. We labeled the patients those required the liver transplant due to the liver decompesation as the high risk group, while those did not

**Table 1. Comparison of clinical characteristics between high risk and low risk groups in patients with LF.**

| Clinical Characteristics | | Low risk group | High risk group |
|---|---|---|---|
| Number of cases | | 174(50%) | 174(50%) |
| Sex | Male | 148(85.1%) | 155(89.1%) |
| | Female | 26(14.9%) | 19(10.9%) |
| Consciousness | Clear awareness | 170(97.8%) | 127(73.0%) |
| | Fuzzy consciousness | 0(0%) | 17(9.8%) |
| | Drowsy | 0(0%) | 12(6.9%) |
| | Sleepy | 0(0%) | 1(0.6%) |
| | Shallow coma | 2(1.1%) | 11(6.3%) |
| | Deep coma | 2(1.1%) | 6(3.4%) |
| Aspartate transaminase (AST) | U/L | 450.26 ± 866.79 | 281.11 ± 549.26 |
| Absolute value of neutrophils (NEUT) | $10^9/L$ | 6.26 ± 5.13 | 6.01 ± 4.36 |
| Lymphocytes Count (LYMPH) | $10^9/L$ | 1.38 ± 0.67 | 1.27 ± 0.73 |
| Creatine (Cr) | μmol/L | 90.81 ± 65.90 | 79.63 ± 43.50 |
| Glutamate transaminase (ALT) | U/L | 473.16 ± 687.52 | 335.15 ± 610.61 |
| Albumin (ALB) | g/L | 33.28 ± 4.97 | 35.28 ± 4.08 |
| Total protein (TPROT) | g/L | 62.51 ± 7.42 | 60.60 ± 8.07 |
| Total bilirubin (TBIL) | μmol/L | 268.99 ± 184.31 | 476.16 ± 206.04 |
| Sodium (Na) | mmol/L | 136.41 ± 4.84 | 137.22 ± 5.74 |
| Calcium (Ca) | mmol/L | 2.21 ± 0.17 | 2.38 ± 0.25 |
| Serum pre-albumin (PA) | mg/L | 64.95 ± 44.12 | 56.89 ± 34.80 |
| Prothrombin standardized ratio (INR) | | 1.39 ± 0.15 | 3.28 ± 1.54 |
| Total cholesterol (CHOL) | mmol/L | 3.57 ± 1.22 | 2.58 ± 1.59 |
| Triglycerides (TRIG) | mmol/L | 1.71 ± 0.91 | 0.79 ± 0.48 |

Values are Expressed as Mean ± Standard deviation.

require as low risk group. In this study, all the patients in both groups were admitted to hospital with impaired liver function. The ratio of high risk patients to low risk patients is 1 to 1. The details of LF patients whose clinical data were used to generate the machine learning models are shown in S1 Table. Table 2 gives the details of input parameters and characteristics. Specially, the consciousness in this paper includes clear awareness, fuzzy consciousness, drowsy, sleepy, shallow coma and deep coma.

## Feature selection

We verify the reasonability of the above predictors from the clinical experience of professional hepatology doctors. Further, predictors with too many missing values (more than 10% over 348 rows of records) are discarded directly to avoid inaccurate predictions.

- Consciousness. Different liver diseases may lead to the dysfunction of the brain. Liver disease often causes the mild hepatic encephalopathy, which leads to incoherent and unclear consciousness. Simple and convenient consciousness detection method is of great help to the early detection of hepatic encephalopathy.

- Aspartate transaminase (AST). The aspartate transaminase is significantly increased when there is extensive damage to liver essence and poor prognosis. Especially, it suggests that hepatitis lesions are chronic and progressive when serum AST increases and exceeds the ALT continuously.

**Table 2. Description of input parameters characteistics.**

| Input attributes | Description | Type |
|---|---|---|
| Consciousness | Mental confusion and sleepiness | Text |
| AST | Aspartate transaminase | Numeric |
| NEUT | Absolute value of neutrophils | Numeric |
| LYMPH | Lymphocytes Count | Numeric |
| Cr | Creatine | Numeric |
| ALT | Glutamate transaminase | Numeric |
| ALB | Albumin | Numeric |
| TPROT | Total protein | Numeric |
| TBIL | Total bilirubin | Numeric |
| Na | Sodium | Numeric |
| Ca | Calcium | Numeric |
| PA | Serum pre-albumin | Numeric |
| INR | Standardized ratio of clotting enzymes | Numeric |
| CHOL | Total Cholesterol | Numeric |
| TRIG | Triglycerides | Numeric |

Table notes the details of input parameters and characteristics.

- Absolute value of neutrophils (NEUT). NEUT is one fraction of the blood cells in the blood. The changes of NEUT can be used to assess infectious diseases, blood system diseases, mononuclear-macrophage system functional advances, and other diseases.

- Lymphocytes Count (LYMPH). Lymphocyte is a type of white blood cell and the smallest white blood cell. It is produced by lymphatic organs, mainly in the lymphatic fluid circulating in the lymphatic tube, and is an important cellular component of the body's immune response function. Lymphocyte is an important indicator of infectious diseases.

- Creatine (Cr). Creatinine in blood comes from exogenous and endogenous, exogenous creatinine is the product of meat food metabolism in the body, endogenous creatinine is the product of muscle tissue metabolism in the body. Creatine concentration can reflect the filter function of the glomerular.

- Glutamate transaminase (ALT). In the acute stage of viral hepatitis, drug toxic liver cell necrosis, ALT is released into the blood in large quantities, so it is an important index for diagnosing viral hepatitis and toxic hepatitis. As long as 1% of liver cells are necrosis, enzyme activity in the blood can be increased by a quarter of a change, so transaminase (especially ALT) is a sensitive indicator of acute liver cell damage.

- Albumin (ALB). The decline of albumin has important clinical significance to reflect the degree of liver function at the time of cirrhosis.

- Total protein (TPROT). Serum total protein (TP), which can be divided into albumin and globulin, has important physiological functions in the body, and the determination of serum total protein is one of the important items of clinical biochemical testing.

- Total bilirubin (TBIL). Total bilirubin, TBil is the sum of direct bilirubin and indirect bilirubin. The liver plays an important role in the metabolism of bilirubin, including the intake, binding and excretion of bilirubin in the blood by liver cells. Obstacles in any of these processes can cause bilirubin to accumulate in the blood and jaundice.

- Sodium (Na). Cirrhosis of the liver, often with hyponatremia, may be related to repeated abdominal water, or commond diuretics, cirrhosis patients often have elevated plasma sodium levels, may be another factor causing the reduction of serum sodium.

- Calcium (Ca). Blood calcium levels are associated with many important functions of the human body. Parathyroid, calcitonin and cholecalcetol play an important role in maintaining normal blood calcium concentrations. Ca represents the concentration of total calcium.

- Serum pre-albumin (PA). PA is a glycoprotein synthesized by the liver. Serum pre-albumin assays can reflect the function of the liver in synthesizing and secreting proteins, and can be used as an early indicator of liver function damage and suggest changes and prognosis of some diseases.

- Standardized ratio of clotting enzymes (INR). Coagulation enzymes are synthesized in the liver. Severe liver disease can lead to clotting enzyme synthesis disorder, which can lead to reduced activity of clotting enzymes, which eventually leads to bleeding.

- Total Cholesterol (CHOL). Total cholesterol includes free cholesterol and cholesterol esters, which are synthesized and stored through the liver. It is an important indicator of liver disease.

- Triglycerides (TRIG). Triglycerides are made from food fats and liver, and are fat molecules formed by long-chain fatty acids and glycerin. It is the most important type of blood lipids.

## Mode selection

Four machine learning models were constructed to determine the incremental discriminatory and reclassification performance of the selected predictors in identifying the exacerbation risk with LF.

(1) Artificial neural network (ANN).

ANN as non-linear statistical data modeling tools are employed to to find patterns between input vector $x$ and outputs $y$. ANN Eq (2) is expressed as:

$$\hat{y} = f(C\sum_{i=1}^{n} w_i x_i - \theta) \tag{2}$$

where $y$ is the output vector, $x$ is the input vector, $w$ is the weight vector, $\theta$ is the natural threshold, $f$ is the activation function both of which are determined only by the training samples.

(2) Classification and regression tree (CART).

CART uses the Gini index (Gini) Eq (3) to select the optimal feature and determines the optimal two-value cut point of the predictor. The Gini Eq (3) is defined as:

$$Gini(p) = \sum_{k=1}^{n} p_k(1 - p_k) = 1 - \sum_{k=1}^{n} p_k^2, \tag{3}$$

where $p_k$ is probability that the sample belongs to class k, $k$ is the number of classes.

For a given sample set D, the Gini Eq (4) is defined:

$$Gini(D) = 1 - \sum_{k=1}^{n} \frac{|C_k|}{|D|}, \tag{4}$$

where $C_k$ is a subset of the sample of class K in D.

Sample set D is split into two parts based on a possible value of feature A Eq (5):

$$D_1 = (x, y) \in D|A(x) = a, D_2 = D - D_1, \tag{5}$$

where $C_k$ is a subset of the sample of class K in D.

Under the condition of feature A, the Gini of set D Eq (6) is defined as:

$$Gini(D, A) = \frac{|D_1|}{|D|} Gini(D_1) + \frac{|D_2|}{|D|} Gini(D_2), \tag{6}$$

where $Gini(D)$ represents the uncertainty of data set $D$, $Gini(D, A)$ denotes the uncertainty of collection D after segmentation $A = a$. The larger the Gini, the greater the uncertainty of the sample set.

(3) Generalized linear model (GLMs).

GLMs is a nonlinear mapping relationship from the input space to the output space, which is proposed to overcome the strict linear hypothesis of common linear model (CLMs) for the input data $D_t$. In a binary classification system, CLMs Eq (7) can be expressed as:

$$f(\hat{y}) = w_1 x_1 + w_2 x_2 + \ldots + w_k x_k + b = w^T x + b, \tag{7}$$

By the anti-function calculation of the contact function $f^{-1}$, the generalized linear model Eq (7) can be obtained Eq (8).

$$\hat{y} = f^{-1}(w^T x + b). \tag{8}$$

where $\hat{y}$ is the output vector, $x$ is the input vector, $w$ is the weight vector, $T$ denotes the operation of the vector transpose, $b$ represents the bias. $f$ denotes the activation function which is determined by the training samples.

(4) Support vector machine (SVM).

SVM constructs a hyperplane in a high dimensional space, which can be used for classification, regression, or other tasks. The solving process is as follows Eq (9):

$$min \quad \frac{1}{2}||(\mathbf{w})||^2 + C\sum_{i=1}^{M}\varepsilon_i s.t. \ y_i[\mathbf{w}.\mathbf{K}(x_i, x_j) + b] + \varepsilon_i \geq 1, \ for \ i = 1, ..., M. \tag{9}$$

where $w$ is the weight vector and b is the bias, both of which are determined only by the training samples. The regular parameter $C$ is a penalty factor, which can balance the model complexity and empirical risk. In addition, $\varepsilon_i$ is positive parameters called slack variables, which represents the distance between the misclassified sample and the optimal hyperplane. Function $\mathbf{K}(x_i, x_j)$ is the Sigmoid kernel.

## Results

(1)Baseline Characteristics of Patients and Clinical Indicators.

A total of 348 LF patients were included from May 2011 to March 2018 retrospectively in this study; We labeled the patients those required the liver transplant due to the liver decompesation as the high risk group, while those did not require as low risk group. In this study, all the patients in both groups were admitted to hospital with impaired liver function. Fifty percent of the patients presented with liver transplant. Mean values of INR, Cr and TBIL for the high risk group were 3.28 ± 1.54 and 79.63 ± 43.50 and 476.16 ± 206.04, respectively (Table 1). While those for the low risk group were 1.39 ± 0.15, 90.81 ± 65.90 and 268.99 ± 184.31, respectively (Table 1).

(2)Comparison of the Classification Performance of Each Model.

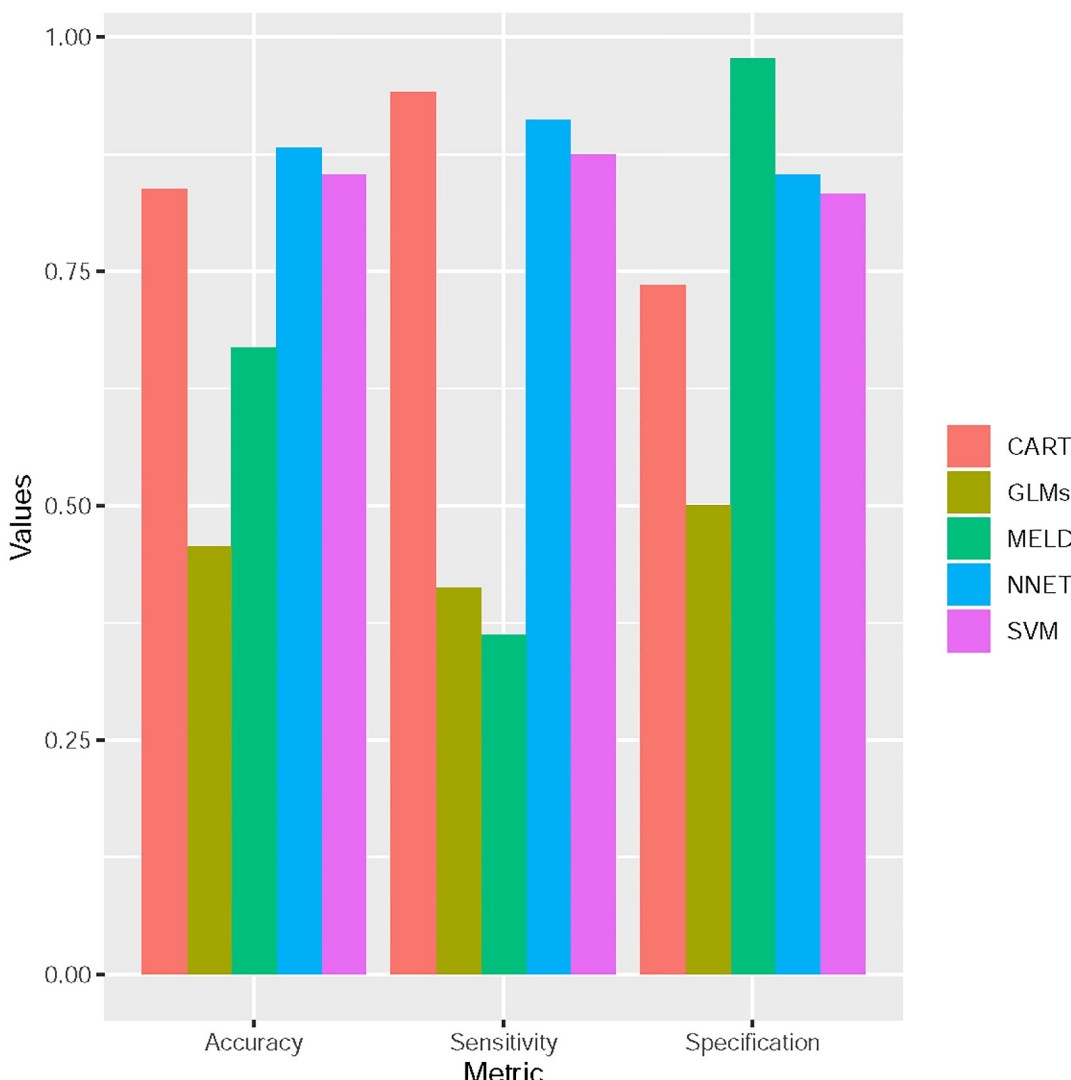

**Fig 1. The comparison of classification ability of predictive models.** the classification ability for the models from model 1 to model 5 was compared.

The classification performance of each model was assessed by the Confusion Matrix. Based on the confusion matrix, we calculated the classification accuracy, sensitivity and specificity. The prediction model was validated using 3-fold cross-validation. Using this method, the classification ability for the models from model 1 to model 5 was compared (Fig 1).

Compared to MELD, the classification ability ability were improved. The overall accuracy of the proposed CART classifier is 79.41% with 95% Confidence Interval (CI):(0.6788, 0.8826). The ANN classifier achieved the best classification performance among the five models. While MELD model showed the 67%.

In addition, Gini index (the model purity assessment) was used to investigate the relative importance of each predictor. Predictors with a smaller Gini value contribute more information to prediction models than ones with higher Gini value (Fig 2).

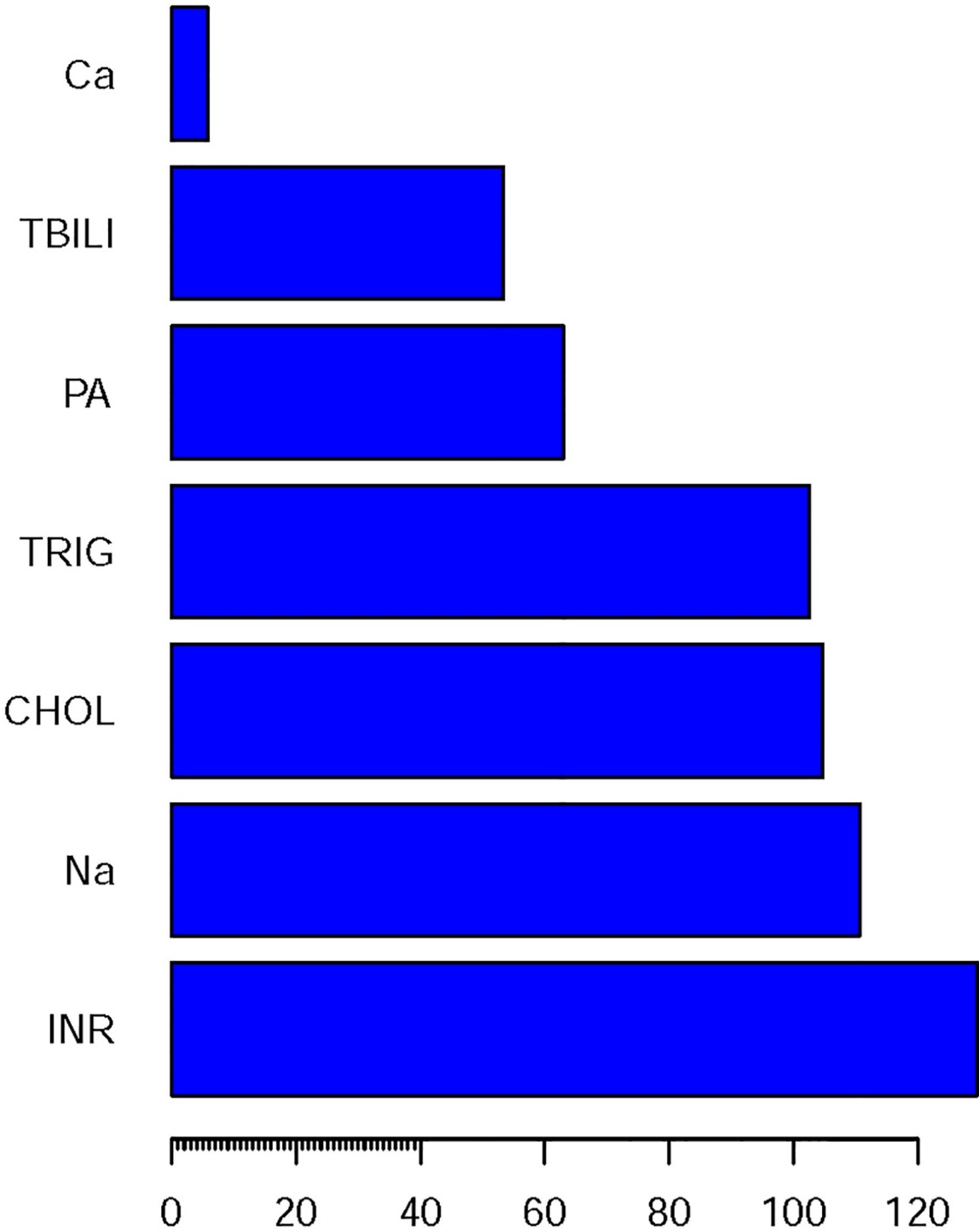

**Fig 2. Importance of each predictor of the CART models.** Predictors with a smaller Gini value contribute more information to prediction models than ones with higher Gini value.

## Discussion

The current study investigated the utility of machine learning and MELD method in the identification of exacerbation risk in patients with liver disease. The main findings are as follows.

First, comprehensive assessment with consciousness, Cr, NEUT, ALT, TBIL, Na, PA, INR and TC, etc, improved the classification ability in the identification of exacerbation risk in patients with liver disease compared with MELD model. Second, the predictors such as INR, NEUT TBIL and PA, etc, seemed to show more important clinical significance than the other predictors.

Previous studies showed that MELD model is a classic model for predicting liver failure. Nevertheless, the assessment of liver failure based only MELD has some limitations, as the LF disease have a heterogeneous natural history and only a small proportion of them actually cause clinical events.

There are some limits in our research. First, the present study used conscience, AST, NEUT, LYMPH, CREAT, ALT, ALB, TPROT, TBILI, Na, Ca, PA, INR, CHOL and TRIG, for the prediction of LF risk. However, the present study retrospectively collected liver disease data after hospitalization event, and the number of patients was relatively small. In order to overcome the chance of overfitting, we used 3-fold cross-validation on the data set to make the model more general. Based on the current study results, approximately 2,000 LF patients would be needed to conclusively validate the results.

## Conclusion

Machine learning method enhanced the identification of exacerbation risk in patients with liver disease that subsequently caused liver failure. The integration of machine learning assessments may improve the identification of exacerbation risk for LF in the future.

## Supporting information

**S1 Table. Liver disease relevant data.** Viral liver disease relevant data underlying the findings described in manuscript.
(XLSX)

## Acknowledgments

We thank Jiayuan Chen for helping with data collection and pre-processing.

## Author Contributions

**Conceptualization:** Junfeng Peng, Mi Zhou.

**Data curation:** Junfeng Peng.

**Software:** Junfeng Peng.

**Supervision:** Ching-Hsing Luo.

**Writing – original draft:** Junfeng Peng, Mi Zhou.

**Writing – review & editing:** Chuan Chen, Xiaohua Xie.

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
