## [Decision Letter · Decision Letter 0]

10 Mar 2020

PONE-D-20-04018

Identification of Exacerbation Risk in Patients with Viral Liver Disease Using Machine Learning Algorithms

PLOS ONE

Dear Dr. Luo,

Thank you for submitting your manuscript to PLOS ONE. After careful consideration, we feel that it has merit but does not fully meet PLOS ONE’s publication criteria as it currently stands. Therefore, we invite you to submit a revised version of the manuscript that addresses the points raised during the review process.

We would appreciate receiving your revised manuscript by Apr 24 2020 11:59PM. To enhance the reproducibility of your results, we recommend that if applicable you deposit your laboratory protocols in protocols.io, where a protocol can be assigned its own identifier (DOI) such that it can be cited independently in the future. For instructions see: http://journals.plos.org/plosone/s/submission-guidelines#loc-laboratory-protocols

We look forward to receiving your revised manuscript.

Kind regards,

Ming-Lung Yu, MD, PhD

Academic Editor

PLOS ONE

Journal Requirements:

"This work was supported by Sun Yat-sen University, China, under Scientific Initiation Project No.67000-18821109 for High-level Experts. No competing financial interests exist."

Reviewers' comments:

Reviewer's Responses to Questions

**Comments to the Author**

1. Is the manuscript technically sound, and do the data support the conclusions?

Reviewer #1: No

Reviewer #2: Partly

2. Has the statistical analysis been performed appropriately and rigorously? 

Reviewer #1: I Don't Know

Reviewer #2: I Don't Know

3. Have the authors made all data underlying the findings in their manuscript fully available?

Reviewer #1: No

Reviewer #2: Yes

4. Is the manuscript presented in an intelligible fashion and written in standard English?

Reviewer #1: Yes

Reviewer #2: Yes

5. Review Comments to the Author

Reviewer #1: Congratulations for the birlliant idea and excellent work for implication of machine learning to help caring patient with liver disease. My personal opinions are as following.

1. Generally speaking, the criteria for ICU admission were shock, respiratory failure or any major organ failure.

What's the exact criteria for admission to intensive care unit, i.e. the criteria for high risk group. Were all the high risk group patients diagnosed of liver decompesation?

2. In Table 1, consciousness was categorized as clear-minded and "Fuzzy". What's the definition of fuzzy conscousness? Does it mean irritable, agitated, hypersomnance, stupor or comatose? GCS?

3. In Table 1, Creatine(Cr) was included as a parameter, however, "patients with chronic kidney disease was excluded" according to the context. Is there specific definition, such as eGFR or MDRD for staging chronic kidney disease, and by which stage the patients were excluded?

4. In Table 1, Prothrombin standardized ratio (INR) 2.6 1 was defined as low risk group. However, Child Pugh Score, a widely accepted scoring system for evaluation the liver status of cirrhotic patients, gives 2 ppoints for INR > 1.7.

5. The goal of the current study was to forcast whether a patient would deteriorate after admission to the hospital. Defining "deteriorate", did the study focus only on liver function deterioration? As we know, a large proportion of patients with end staged liver disease sufferred from deteriorated condition due to events of renal failure, such as hepatorenal syndrome, or sepsis, such as SBP, or hypovolemic shock due to recurrent UGI bleeding (variceal bleeding), how did the study stratify different types of deterioration of liver function, is the etiology of deterioration considered?

6. Quoted from paragraph of Discussion: "Nevertheless,the assessment of liver failure based only MELD has some limitations, as the LD disease have a heterogeneous natural history and only a small 141 proportion of them actually cause clinical events." The current study did not stress some of the the well known features of VLD, such as HBV DNA level, HBsAg level or HBeAg/HBeAb for CHB flare. These established features of VLD were considered to play important role of liver function deterioration, and may be more predictive than some of the parameters as NEUT or TC.

7. Is it possible to share, at least part of the raw data and calculation of CART classifier? Then we can discuss the pitfalls of machine learning implication for liver disease.

Thank you.

Reviewer #2: Peng et al. conducted a retrospective study to investigate the risk for patients with viral liver disease (VLD) who would deteriorate after admission to the hospital. A total of 308 liver dysfunction (LD) patients were identified, and among them, 283 VLD patients were included for analysis. The outcomes of VLD patients were dichotomously divided as high and low risk groups, and the high risk group was defined as those who required ICU care. They built the machine learning model by incorporating 9 key clinical indicators. Four different machine learning algorithms were utilized and their performance was compared to MELD scoring system. Finally, they claimed the performance of the proposed machine learning model is superior to the MELD model. Although machine learning is an interesting approach to help improve the risk stratification for VLD patients, some caveats of this study need to be carefully addressed in order to improve the quality.

1. The major issue is the definition of clinical outcomes (high and low risk groups) that were only based on the need of ICU care. It is uncertain whether the need of ICU care was due to liver-related causes or not. This needs to be clarified. Presumably, liver-related mortality or hepatic failure/the need for liver transplantation may be a more appropriate outcome for this analysis.

2. Another issue is the choice of the 9 clinical indicators. There are quite a few number of clinical parameters associated with patients’ outcomes. What is the rationale for choosing only these 9 parameters. This should be described.

3. As raised by the authors, no validation cohort is a concerning issue. Because this study has a small case number and 9 parameters for analysis, the chance of overfitting cannot be ignored.

4. Based on the title of this paper, all viral hepatitis patients should be included. Why were only patients with hepatitis B and/or hepatitis E included? Hepatitis C should be also a common viral hepatitis. Was there any reason to exclude hepatitis C from this analysis?

5. Whether patients have underlying liver disease is also an important factor for the severity of hepatitis episode. Was the hepatitis B in these patients acute or chronic? In addition, was advanced fibrosis or cirrhosis present before this episode of hepatitis?

6. What is the timing for collecting the 9 clinical indicators? Were they all collected upon admission? This may affect the results of this analysis.

7. In this study, there were a total of 308 LD patients, whose clinical characteristics are shown in Table 1. However, it seemed that only 283 VLD patients were included for analysis. If this is the case, they should show the clinical characteristics of these 283 patients.

8. In Table 2, ‘Na’ is missing.

6. PLOS authors have the option to publish the peer review history of their article (what does this mean?). If published, this will include your full peer review and any attached files.

Reviewer #1: Yes: Ta-Wei Liu

Reviewer #2: No

---

## [Author Response · Author response to Decision Letter 0]

20 Jun 2020

Dear Editors and Reviewers:

Thank you for your letter and for the reviewers’ comments concerning our manuscript entitled “Identification of Exacerbation Risk in Patients with Liver Dysfunction Using Machine Learning Algorithms” (PONE-D-20-04018). Those comments are all valuable and very helpful for revising and improving our paper, as well as the important guiding significance to our researches. We have studied comments carefully and have made correction which we hope meet with approval. The main corrections in the paper and the responds to the reviewer’s comments are as flowing:

Reviewer #1: Congratulations for the birlliant idea and excellent work for implication of machine learning to help caring patient with liver disease. My personal opinions are as following.

1. Generally speaking, the criteria for ICU admission were shock, respiratory failure or any major organ failure. What's the exact criteria for admission to intensive care unit, i.e. the criteria for high risk group. Were all the high risk group patients diagnosed of liver decompesation?

Reviewer #1-RESPONSE: Special thanks to you for your good comments. Inspired by your comments, we realized that there was a problem with study population selection. Thus, we resampled the research group. We labeled the patients those required the liver transplant due to the liver decompesation as the high risk group , while those did not require as low risk group. In this study, all the patients in both groups were admitted to hospital with impaired liver function.

2. In Table 1, consciousness was categorized as clear-minded and "Fuzzy". What's the definition of fuzzy conscousness? Does it mean irritable, agitated, hypersomnance, stupor or comatose? GCS?

Reviewer #1-RESPONSE: The state of consciousness includes clear awareness, fuzzy consciousness, drowsy, sleepy, shallow coma and deep coma . We have added the corresponding description in the second part of the paper.

3. In Table 1, Creatine(Cr) was included as a parameter, however, "patients with chronic kidney disease was excluded" according to the context. Is there specific definition, such as eGFR or MDRD for staging chronic kidney disease, and by which stage the patients were excluded?

Reviewer #1-RESPONSE: eGFR-MDRD is an indicator of chronic kidney disease in serum testing. The value of eGFR-MDRD can be calculated by weight, creatinine, and age. Serum creatinine is an important indicator of kidney function. In the new data collection, we do not exclude the patients with chronic kidney disease. We focused the patients those required the liver transplant due to the liver decompesation as the study population.

4. In Table 1, Prothrombin standardized ratio (INR) 2.6 1 was defined as low risk group. However, Child Pugh Score, a widely accepted scoring system for evaluation the liver status of cirrhotic patients, gives 2 ppoints for INR > 1.7.

Reviewer #1-RESPONSE: We are very sorry for our negligence that we made a mistake to select the patients those require ICU admission as the high risk group. We resampled the study population selection and select the patients those required the liver transplant due to the liver decompesation as the high risk group, while those did not require as low risk group. In this study, all the patients in both groups were admitted to hospital with impaired liver function. In the new collection, the INR in the low risk group was 1.39.

5. The goal of the current study was to forcast whether a patient would deteriorate after admission to the hospital. Defining "deteriorate", did the study focus only on liver function deterioration? As we know, a large proportion of patients with end staged liver disease sufferred from deteriorated condition due to events of renal failure, such as hepatorenal syndrome, or sepsis, such as SBP, or hypovolemic shock due to recurrent UGI bleeding (variceal bleeding), how did the study stratify different types of deterioration of liver function, is the etiology of deterioration considered?

Reviewer #1-RESPONSE: 

As the reviewer described there were different types of deterioration of liver function. The complex deteriorated condition with end staged liver disease often exists due to the events of renal failure, such as hepatorenal syndrome, or sepsis, such as SBP, or hypovolemic shock due to recurrent UGI bleeding (variceal bleeding). Inspired by the reviewers’ insights, we resampled the study population by whether the patient has had a liver transplant with end staged liver disease as the high risk group, while those did not require as the low risk group. In fact, there are multiple severity ratings, and for practical operability, we identify classification as two categories (high risk and low risk)

6. Quoted from paragraph of Discussion: "Nevertheless,the assessment of liver failure based only MELD has some limitations, as the LD disease have a heterogeneous natural history and only a small 141 proportion of them actually cause clinical events." The current study did not stress some of the the well known features of VLD, such as HBV DNA level, HBsAg level or HBeAg/HBeAb for CHB flare. These established features of VLD were considered to play important role of liver function deterioration, and may be more predictive than some of the parameters as NEUT or TC.

Reviewer #1-RESPONSE: We checked the well known features of VLD in the raw data. We found that there were a lot missing values for the well known features such as HBsAg, HBeAg/HBeAb, HBV DNA level. The reason for missing values for the well known features may be that tests occurred before the current data, or for the other reasons. Considering the Reviewer’s profound opinion, we were concerned about the acute deterioration risk prediction in patients with the end staged liver disease caused by a variety of causes (viral liver, alcoholic liver, medicinal liver, pregnancy liver, etc.) using the common clinical indicators using machine learning methods.

7. Is it possible to share, at least part of the raw data and calculation of CART classifier? Then we can discuss the pitfalls of machine learning implication for liver disease.

Reviewer #1-RESPONSE: Thank you very much again for your profound comments, which have been of great help to our research work, and we will show you the entire desensitization data and procedures.

Thank you.

Reviewer #2: Peng et al. conducted a retrospective study to investigate the risk for patients with viral liver disease (VLD) who would deteriorate after admission to the hospital. A total of 308 liver dysfunction (LD) patients were identified, and among them, 283 VLD patients were included for analysis. The outcomes of VLD patients were dichotomously divided as high and low risk groups, and the high risk group was defined as those who required ICU care. They built the machine learning model by incorporating 9 key clinical indicators. Four different machine learning algorithms were utilized and their performance was compared to MELD scoring system. Finally, they claimed the performance of the proposed machine learning model is superior to the MELD model. Although machine learning is an interesting approach to help improve the risk stratification for VLD patients, some caveats of this study need to be carefully addressed in order to improve the quality.

1. The major issue is the definition of clinical outcomes (high and low risk groups) that were only based on the need of ICU care. It is uncertain whether the need of ICU care was due to liver-related causes or not. This needs to be clarified. Presumably, liver-related mortality or hepatic failure/the need for liver transplantation may be a more appropriate outcome for this analysis.

Reviewer #2-RESPONSE: Special thanks to you for your good comments. Inspired by your comments, we realized that there was a problem with study population selection. Thus, we resampled the research group. We labeled the patients those required the liver transplant due to the liver decompesation as the high risk group , while those did not require as low risk group. In this study, all the patients in both groups were admitted to hospital with impaired liver function.

2. Another issue is the choice of the 9 clinical indicators. There are quite a few number of clinical parameters associated with patients’ outcomes. What is the rationale for choosing only these 9 parameters. This should be described.

Reviewer #2-RESPONSE: Considering the Reviewer’s suggestion, we have added the rationale for choosing the clinical indicators. The principle of clinical indicators choice is to select the indicators with less missing values, and to select indicators with clinical significance among them. 

3. As raised by the authors, no validation cohort is a concerning issue. Because this study has a small case number and 9 parameters for analysis, the chance of overfitting cannot be ignored.

Reviewer #2-RESPONSE: In order to overcome the chance of overfitting, we used 10 fold cross validation on the data set to make the model more general. In the future, we will collect more data for model improvement.

4. Based on the title of this paper, all viral hepatitis patients should be included. Why were only patients with hepatitis B and/or hepatitis E included? Hepatitis C should be also a common viral hepatitis. Was there any reason to exclude hepatitis C from this analysis?

Reviewer #2-RESPONSE: We are very sorry for our negligence that we did not state clearly our research objectives. In fact, we intended to predict deterioration risk of liver dysfunction using machine learning methods with the clinical indicators. The causes of liver decompesation include hepatitis B, hepatitis C, hepatitis E, drug-induced liver damage, autoimmune liver disease, liver bean-like nuclear degeneration, alcoholic liver disease, fatty liver disease during pregnancy, etc. We first collected the patients those required the liver transplant due to the liver decompesation as the high risk group , then collected those did not require as low risk group. In the new collection data, the majority of patients are hepatitis B virus patients, and a small number are Hepatitis C virus patients (only 6).

5. Whether patients have underlying liver disease is also an important factor for the severity of hepatitis episode. Was the hepatitis B in these patients acute or chronic? In addition, was advanced fibrosis or cirrhosis present before this episode of hepatitis?

Reviewer #2-RESPONSE: We resampled the research group. The high risk group consisted of the patients those required the liver transplant due to the liver disfunction. The causes of liver disfunction included acute or chronic liver disfunction or the other underlying liver disease.

6. What is the timing for collecting the 9 clinical indicators? Were they all collected upon admission? This may affect the results of this analysis.

Reviewer #2-RESPONSE: We collected the clinical indicators before treatment (liver transplant) after admission.

7. In this study, there were a total of 308 LD patients, whose clinical characteristics are shown in Table 1. However, it seemed that only 283 VLD patients were included for analysis. If this is the case, they should show the clinical characteristics of these 283 patients.

Reviewer #2-RESPONSE: We resampled the research group. The research group inclusion was whether a liver transplant had been performed. We focused on the deterioration risk prediction which caused a liver transplant using the clinical indicators. Thus, we are not concerned about the proportion of VLD patients in the research group.

8. In Table 2, ‘Na’ is missing.

 Reviewer #2-RESPONSE: We have added the description of ‘Na ‘.

---

## [Decision Letter · Decision Letter 1]

20 Jul 2020

PONE-D-20-04018R1

Identification of Exacerbation Risk in Patients with Liver Dysfunction Using Machine Learning Algorithms

PLOS ONE

Dear Dr. Luo,

Thank you for submitting your manuscript to PLOS ONE. After careful consideration, we feel that it has merit but does not fully meet PLOS ONE’s publication criteria as it currently stands. Therefore, we invite you to submit a revised version of the manuscript that addresses the points raised during the review process.

We look forward to receiving your revised manuscript.

Kind regards,

Ming-Lung Yu, MD, PhD

Academic Editor

PLOS ONE

Reviewers' comments:

Reviewer's Responses to Questions

**Comments to the Author**

1. If the authors have adequately addressed your comments raised in a previous round of review and you feel that this manuscript is now acceptable for publication, you may indicate that here to bypass the “Comments to the Author” section, enter your conflict of interest statement in the “Confidential to Editor” section, and submit your "Accept" recommendation.

Reviewer #1: All comments have been addressed

Reviewer #2: All comments have been addressed

2. Is the manuscript technically sound, and do the data support the conclusions?

Reviewer #1: Partly

Reviewer #2: Yes

3. Has the statistical analysis been performed appropriately and rigorously? 

Reviewer #1: I Don't Know

Reviewer #2: Yes

4. Have the authors made all data underlying the findings in their manuscript fully available?

Reviewer #1: No

Reviewer #2: Yes

5. Is the manuscript presented in an intelligible fashion and written in standard English?

Reviewer #1: No

Reviewer #2: Yes

6. Review Comments to the Author

Reviewer #1: 1) Is the paper edited by native speaker of English? There were plenty of wording problems and grammar issues as:

- Grammar and misspelling : "Some liver diseasemay permanent liver 6

damage such as viral liver disease would lead to permanent liver failure"

"accuracyon"

- Missing words? "it is difficult to resolve the onset of the liver with some symptoms or indicators"

- Misspelling? "coagulation assase"

- In the paragraph of Feature Selection: "Conscience.Conscience.Different liver diseases..."

In terms of hepatoencephalopathy, usually "consciousness", instead of "conscience", is influenced, .

2) In the paragraph of Feature Selection: about "Calcium"

Quote: "maintaining the normal concentration of blood calcium plays an important role in the hormones

are metformin, calcite and cholate calcinol."

As a matter of fact, metformin is an gluconeogenesis suppressor and insulin sensitizer, calcite is a carbonate mineral, and Calcinol is registered name of calcium supplement? It seems metformin, calcite and calcinol did not belong to the category of "hormone".

3) Table 1, lacks the units of each parameter, such as AST, ALT, ALb, TPROT..

What is "Ca", total calcium or ionized calcium.

Glucosatosinase (AST)  should be Aspartate transaminase?

Glutamate transaminase (ALB) -> should be Albumin?

4) In "Model Selection" (3) Generalized linear model (GLMs),

f(y) = w^Tx + b;

The linear predictor was "w^Tx". The multiplier of x in the predictor was usually set to be constant,

please indicate the role of the parameter "T" in the equation.

By the way, the equation adopt "b" instead of theat as a natural threshold.

The 15 indicators were continuos variables (except for consciousness), the GLMs which could be applied are:

linear regression, ANCOVA, Poisson Regression or Multinomial response. Which GLM was choosed in this study?

Actually the equation f(y) = w^Tx + b seems to describe the hyperplane of SVM.

5) Just curious, the authors used the CART to evaluate the 15 clinical indicators to idendify risk of liver failure, and 10 fold cross validation was used to prevent overfitting, why Random Forest was not applied in the first place?

6) In "Discussion", quote "There are some limits in our research. First, the present study used consciousness, 208

Cr, NEUT,ALT, TBIL, Na ,PA,INR and TC, etc, for the prediction of LF risk.".

There were only 9 clinical indicators mentioned:　consciousness, Cr, NEUT,ALT, TBIL, Na ,PA,INR and TC.

Were the other 6 indicators (AST, lymphocyte count, albumin, total protein, calcium and triglyceride) excluded and why?

Reviewer #2: The authors have already satisfactorily addressed all of the concerns I raised. I have no more comments on this manuscript.

7. PLOS authors have the option to publish the peer review history of their article (what does this mean?). If published, this will include your full peer review and any attached files.

Reviewer #1: No

Reviewer #2: No

---

## [Author Response · Author response to Decision Letter 1]

17 Aug 2020

Reviewers' comments:

Reviewer's Responses to Questions

Comments to the Author

1. If the authors have adequately addressed your comments raised in a previous round of review and you feel that this manuscript is now acceptable for publication, you may indicate that here to bypass the “Comments to the Author” section, enter your conflict of interest statement in the “Confidential to Editor” section, and submit your "Accept" recommendation.

Reviewer #1: All comments have been addressed

Reviewer #2: All comments have been addressed

2. Is the manuscript technically sound, and do the data support the conclusions?

Reviewer #1: Partly

Reviewer #2: Yes

3. Has the statistical analysis been performed appropriately and rigorously? 

Reviewer #1: I Don't Know

Reviewer #2: Yes

4. Have the authors made all data underlying the findings in their manuscript fully available?

Reviewer #1: No

Reviewer #2: Yes

5. Is the manuscript presented in an intelligible fashion and written in standard English?

Reviewer #1: No

Reviewer #2: Yes

6. Review Comments to the Author

Reviewer #1: 1) Is the paper edited by native speaker of English? There were plenty of wording problems and grammar issues as:

- Grammar and misspelling : "Some liver diseasemay permanent liver 6

damage such as viral liver disease would lead to permanent liver failure"

"accuracyon"

- Missing words? "it is difficult to resolve the onset of the liver with some symptoms or indicators"

- Misspelling? "coagulation assase"

- In the paragraph of Feature Selection: "Conscience.Conscience.Different liver diseases..."

In terms of hepatoencephalopathy, usually "consciousness", instead of "conscience", is influenced, .

Reviewer #1-RESPONSE: Special thanks to you for your careful examination of our manuscripts. Inspired by your comments, we have checked the spelling and grammar of the whole paper carefully. 

- Grammar and misspelling :

 "Some liver diseasemay permanent liver 6 damage such as viral liver disease would lead to permanent liver failure" has been modified to “Some liver disease such as viral hepatitis would lead to permanent liver failure”.

"accuracyon" has been modified to "accuracy on"

 - Missing words? 

"it is difficult to resolve the onset of the liver with some symptoms or indicators" has been modified to “it is difficult to identify the onset of liver disease deterioration with some clinical symptoms or indicators”.

- Misspelling? 

"coagulation assase" has been modified to “international normalized prothrombin ratio”.

- In the paragraph of Feature Selection: "Conscience.Conscience.Different liver diseases..."

In terms of hepatoencephalopathy, usually "consciousness", instead of "conscience", is influenced. “conscience” has been modified to "consciousness".

2) In the paragraph of Feature Selection: about "Calcium"

Quote: "maintaining the normal concentration of blood calcium plays an important role in the hormones are metformin, calcite and cholate calcinol."

As a matter of fact, metformin is an gluconeogenesis suppressor and insulin sensitizer, calcite is a carbonate mineral, and Calcinol is registered name of calcium supplement? It seems metformin, calcite and calcinol did not belong to the category of "hormone".

Reviewer #1-RESPONSE: We have revised the inappropriate statement “the hormones are metformin, calcite and cholate calcinol” to “Blood calcium levels are associated with many important functions of the human body. Parathyroid, calcitonin and cholecalcetol play an important role in maintaining normal blood calcium concentrations.”

3) Table 1, lacks the units of each parameter, such as AST, ALT, ALb, TPROT. What is "Ca", total calcium or ionized calcium. Glucosatosinase (AST)  should be Aspartate transaminase? Glutamate transaminase (ALB) -> should be Albumin?

Reviewer #1-RESPONSE: We have the units of each parameter. "Ca" represents the total calcium and we have added the description in the feature selection section. Glucosatosinase (AST) has been modified to “Aspartate transaminase”. “Glutamate transaminase (ALB)” has been modified to “Albumin (ALB)”

4) In "Model Selection" (3) Generalized linear model (GLMs), f(y) = w^Tx + b;

The linear predictor was "w^Tx". The multiplier of x in the predictor was usually set to be constant, please indicate the role of the parameter "T" in the equation. By the way, the equation adopt "b" instead of theat as a natural threshold. The 15 indicators were continuos variables (except for consciousness), the GLMs which could be applied are: linear regression, ANCOVA, Poisson Regression or Multinomial response. Which GLM was choosed in this study? Actually the equation f(y) = w^Tx + b seems to describe the hyperplane of SVM.

Reviewer #1-RESPONSE: We have modified the description of Generalized linear model (GLMs) to “w is the weight vector, T denotes the operation of the vector transposition ,b represents the bias. f denotes the activation function which is determined by the training samples”. We checked and ran our procedures and found that binomial family was used as the contact function and the binomial family the links logit, probit, cauchit, (corresponding to logistic, normal and Cauchy CDFs respectively) log and cloglog (complementary log-log), thus, GLMs was applied as a logistic regression model. We are very sorry for our negligence that we did not state clearly the equation f(y) = w^Tx + b representing the common linear model.

5) Just curious, the authors used the CART to evaluate the 15 clinical indicators to idendify risk of liver failure, and 10 fold cross validation was used to prevent overfitting, why Random Forest was not applied in the first place?

Reviewer #1-RESPONSE: Random forest is generated by the ensemble of decision trees. It is widely used in the analysis and modeling of medical scenarios due to its rapidity, high accuracy, and robustness. However, real-world medical scenes usually require models with interpretability, random forest as a black box model can not meet the interpretability requirement. While the CART model can meet the interpretability requirement and achieve relatively high prediction accuracy by prevent overfitting. This is the reason why Random Forest was not applied in the first place.

6) In "Discussion", quote "There are some limits in our research. First, the present study used consciousness, 208 Cr, NEUT,ALT, TBIL, Na ,PA,INR and TC, etc, for the prediction of LF risk.". There were only 9 clinical indicators mentioned:　consciousness, Cr, NEUT,ALT, TBIL, Na ,PA,INR and TC. Were the other 6 indicators (AST, lymphocyte count, albumin, total protein, calcium and triglyceride) excluded and why?

Reviewer #1-RESPONSE: We are very sorry for our negligence that we did not state clearly the clinical indicators used in our research. In fact , we used the 15 clinical indicators in our research. We have modified the “First, the present study used consciousness, Cr, NEUT,ALT, TBIL, Na ,PA,INR and TC, etc, for the prediction of LF risk.” to “There are some limits in our research. First, the present study used conscience, AST, NEUT, LYMPH, CREAT, ALT, ALB, TPROT, TBILI, Na, Ca, PA, INR, CHOL and TRIG, for the prediction of LF risk”. Thank you very much again for your good comments, which have been of great help to our research work.

Reviewer #2: The authors have already satisfactorily addressed all of the concerns I raised. I have no more comments on this manuscript.

Thank you very much again for your good comments, which have been of great help to our research work.

7. PLOS authors have the option to publish the peer review history of their article (what does this mean?). If published, this will include your full peer review and any attached files.

Do you want your identity to be public for this peer review? For information about this choice, including consent withdrawal, please see our Privacy Policy.

Reviewer #1: No

Reviewer #2: No

---

## [Decision Letter · Decision Letter 2]

3 Sep 2020

Identification of Exacerbation Risk in Patients with Liver Dysfunction Using Machine Learning Algorithms

PONE-D-20-04018R2

Dear Dr. Luo,

We’re pleased to inform you that your manuscript has been judged scientifically suitable for publication and will be formally accepted for publication once it meets all outstanding technical requirements.

Kind regards,

Ming-Lung Yu, MD, PhD

Academic Editor

PLOS ONE

Additional Editor Comments (optional):

Reviewers' comments:

Reviewer's Responses to Questions

**Comments to the Author**

1. If the authors have adequately addressed your comments raised in a previous round of review and you feel that this manuscript is now acceptable for publication, you may indicate that here to bypass the “Comments to the Author” section, enter your conflict of interest statement in the “Confidential to Editor” section, and submit your "Accept" recommendation.

Reviewer #1: All comments have been addressed

Reviewer #2: All comments have been addressed

2. Is the manuscript technically sound, and do the data support the conclusions?

Reviewer #1: Partly

Reviewer #2: Yes

3. Has the statistical analysis been performed appropriately and rigorously? 

Reviewer #1: I Don't Know

Reviewer #2: Yes

4. Have the authors made all data underlying the findings in their manuscript fully available?

Reviewer #1: Yes

Reviewer #2: Yes

5. Is the manuscript presented in an intelligible fashion and written in standard English?

Reviewer #1: Yes

Reviewer #2: Yes

6. Review Comments to the Author

Reviewer #1: The authors had tried to answer the questions and comments raised by the reviewers. Publication of the paper may encourage further research implementing machine learning to evaluate and predict clinical liver disease.

Reviewer #2: (No Response)

7. PLOS authors have the option to publish the peer review history of their article (what does this mean?). If published, this will include your full peer review and any attached files.

Reviewer #1: No

Reviewer #2: No

---

## [Editor Report · Acceptance letter]

7 Sep 2020

PONE-D-20-04018R2 

Identification of Exacerbation Risk in Patients with Liver Dysfunction Using Machine Learning Algorithms 

Dear Dr. Luo:

I'm pleased to inform you that your manuscript has been deemed suitable for publication in PLOS ONE. Congratulations! Your manuscript is now with our production department. 

Kind regards, 

on behalf of

Dr. Ming-Lung Yu 

Academic Editor

PLOS ONE